# The characteristics of spatial expansion and driving forces of land urbanization in counties in central China: A case study of Feixi county in Hefei city

Huafu Jiao[1], Xiangxiang Zhang[1], Chao Yang[1], Xianzhong Cao[2]*

**1** School of Geography and Tourism, Anhui Normal University, Wuhu, Anhui Province, China, **2** The Center for Modern Chinese City Studies, East China Normal University, Shanghai, China

* cao5956@163.com

## Abstract

Urbanization in Central and Western China has attracted increasing attention in the advent of new-type urbanization in China and the age of 'Global Urbanism'. Although land urbanization is at the epicenter of Chinese urbanization, its process and driving forces in counties beyond the Eastern coastal areas are less known. This paper therefore seeks to investigate the spatial expansion and driving forces of land urbanization in Feixi county, a relatively advanced county neighboring Hefei city proper in Anhui province in Central China. Based on the land-use change survey data, remote sensing interpretation data, and statistical year-book data of Feixi county from 2002 to 2016, it is revealed that the overall scale of urban land in Feixi county increased remarkably, though with obvious temporal and geographical variants. The year 2009 appears to be the cutting line between quantity-based land urbanization and quality-based land urbanization. Land urbanization rate is sensitive to both geographical distance and traffic accessibility to Hefei city proper. Driving forces of land urbanization in Feixi county are summarized as: (1) central city expansion, (2) government-led industrial park construction, and (3) large industrial projects along traffic corridors. A better understanding of urbanization in Feixi county cannot be fulfilled without taking into account the wider spatial process in Hefei city and the Yangtze River Delta city-region.

## 1. Introduction

The era of 'Global Urbanism' has been dawning since the onset of the 21[st] century, as a consistently increasing number of the world's population settle in urban areas [1–3]. As it is asserted by the United Nations, 'In 2008 the world reaches an invisible but momentous milestone. For the first time in history, more than half its human population, 3.3 billion people, will be living in urban areas' [4]. In the meantime, the world urbanization process is witnessing 'new geographies' which is characterized as the dislocation of focal urbanizing territories from the Global North to Global South [5–7]. Developing countries in the Global South are not only contributing more dramatically to the world urbanization in quantity and quality but also giving birth to divergent urban experience that is intriguing for both academic and policy inquiries.

**Data Availability Statement:** All relevant data are within the manuscript and its Supporting Information files

**Funding:** National Natural Science Foundation of China (Grant No. 41671163)

**Competing interests:** The authors have declared that no competing interests exist.

China, the world's largest developing country and second largest economy, has been playing a crucial role in shaping the present global 'Urban Age' [1], especially after its economic reform and opening-up in the late 1970s [8–12]. During the early reform period, Chinese urbanization was primarily stimulated by rural industrialization in the countryside where the step-by-step reform initially took place [13,14]. Since the mid-1990s, however, Chinese urbanization has transformed towards one much more featured by land-centred and city-based, which to a greater extent facilitates the urbanization process and enlarges the urbanized geographical scale in China. China's land-centred urbanization is rooted in the land reform initiated in 1987 that generated the paid transfer of land use rights, the tax-sharing system launched in 1994 that upscaled the fiscal power to the central state while kept urban development responsibility downscaled to local authorities, and the housing reform in 1998 that boomed the urban real estate market [15–18]. As Lin et al. [19] rightly point out, 'it is the reshuffling of state power that has been the political origin of phenomenal urbanization and massive land development' in post-reform China. According to the China Construction Statistical Yearbooks during 1991–2017, the urban built-up area in China increased strikingly from 12,855.7 km$^2$ in 1990 to 54,331.5 km$^2$ in 2016, with an average annual growth rate of 5.70%.

Since the 2000s, particularly in the aftermath of the 2008 global financial crisis, the geographical focus of gravity of Chinese urbanization has been shifting [20–22]. This emerging variation is twofold. First, along with the return of migrants and relocation of industries mostly in labor-intensive categories from the coastal to inland areas, as well as the state initiatives to reduce inter-regional disparity and promote more spatially even regional development, central and western China recently has gone through a faster urbanization process and become emerging engines for new waves of Chinese urbanization [12,22,23]. Second, guided by the National New-type Urbanization Plan (2014–2020) as China's first official plan on urbanization issued by the State Council in 2014, city-regionalization and urban-rural integration have been new avenues to sustain Chinese urbanization since the 2010s [24–26]. Under this circumstance, two fresh urbanization trends may be anticipated. On the one hand, central and key nodal cities in economically advanced city-regions will be filled with rising momentum due to improved inter-city infrastructure and coordinated inter-city division of labor [25,27]. On the other hand, counties and towns, especially those neighboring central cities in flourishing city-regions, will shoulder additional responsibilities for population settlement, land expansion, and public service provision [26,28].

Land urbanization was first proposed by Lu [29] in 2006, it means that in a certain country and region, due to the advancement of urbanization, land has been gradually changed to urban. Although this newly emergent Chinese urbanization dynamics is alluring, empirical studies, particularly detailed case-based scrutinization, in contrast to abundant literature on land urbanization in eastern coastal metropolitans [18,19,25,30,31], are insufficiently addressed. Existing studies are inclined to pay attention on spatial-temporal characters and driving forces of Chinese land-centred urbanization, including descriptive analysis on location and rate of land conversion to non-agricultural use at the national level [17,32], and explanatory efforts on its political and institutional underpinnings [12,16], as well as the hybrid mechanisms derived from the interfusion of neoliberal market forces, socialist legacy, and local path-dependent development [19] (Lin et al., 2015). Li et al. [33] point that multiple factors including socioeconomic, physical, proximity, accessibility, and neighborhood factors have driven urban expansion in China; Li et al. [34] explored the spatiotemporal characteristics of Chinese urban expansion and adopted a geographically weighted regression (GWR) method to determine this spatial heterogeneity, The results revealed the spatial heterogeneity in the determinants of urban expansion, marketization was vital for urban expansion and had a stronger impact in the developed eastern and southern regions than in the less-developed

northern and western regions, globalization and decentralization bi-directionally affected urban expansion. Nevertheless, detailed case studies on land urbanization in counties and towns in central and western China are by no means meaningless especially when it comes to the present transition towards a more 'inclusive' and spatially balanced new-type urbanization [24,26,28]. Zooming in on the spatial characters and driving forces of land urbanization in counties and towns in central and western China will offer updated evidence of novel trends in new-type urbanization, and yield valuable insights into divergent urban experience under different sub-national contexts for blooming 'global urban studies' [3]. The United Nations proposed in the 2030 Agenda for Sustainable Development Goals (SDGs), SDG11.3.1 has been used to evaluate land urbanization [35].

An investigation into land urbanization in Feixi county of Hefei city in Anhui province in central China provides a fascinating and timely opportunity to fill in the research gap highlighted above. Anhui province, located in central China with geographical proximity to the Yangtze River, is one of the first two pilot provinces designated for implementing new-type urbanization [26]. Feixi county is a typical representative of county economy in Anhui. In 2009, it was the first county-level administrative unit in Anhui to become one of the economically top 100 counties in China. The amount of permanent population and GDP in 2018 there reached 780,200 and 70.31 billion yuan, respectively [36]. Therefore, this paper attempts to probe into the spatial expansion and driving forces of land urbanization in Feixi county since the early 2000s. Under the support of both quantitative and qualitative methodologies, it aims to unveil emerging land urbanization realities in the specific context of central China, thereby enriching the multicolored image of 'Global Urbanism'.

The remainder of this paper is organized as follows. The next section introduces the materials and methods. Section 3 describes the spatial expansion characteristics of land urbanization in Feixi county. Section 4 analyses the driving forces of land urbanization in Feixi county. The last section concludes the paper.

## 2. Materials and methods

### 2.1 Data source and processing

Hefei, the provincial capital, operates as the undisputed political, economic, and urbanization center in Anhui, as well as a sub-center in the Yangtze River Delta city-region. Feixi county is on the westward of Hefei city proper and under the jurisdiction of Hefei city. Data used in this study mainly includes the land-use change survey data from 2006 to 2016 and the 30 m resolution landsat5 TM remote sensing image in 2002. In addition, the 30 m resolution landsat5 TM in 1995, 2002, 2006, and 2009, and the 30 m resolution landsat8 OLI/TIRS remote sensing image data in 2013 and 2016 are used for radiometric calibration and atmospheric correction. Furthermore, the urban construction land data of Hefei city and its counties (e.g., Feixi, Feidong, and Changfeng), are extracted by the human-computer interactive visual interpretation method. Relevant Statistical Yearbooks of Feixi and Hefei and social and economic development planning and policy documents in different time-periods are sourced as supplementary materials.

Some data issues should be clearly elaborated before the interpretation of methodologies. Since the 2006 land use survey data was updated on the basis of the first national land survey data and the 2009–2016 land-use change survey data was updated on the basis of the second national land survey data, the comparability of them is poor. Therefore, based on the survey data of land-use change in Feixi county in 2009, this study uses landsat5 TM remote sensing image interpretation data from 2006 to adjust the land-use change survey data of Feixi county in 2006. Moreover, Feixi county carried out two key administrative division adjustments in

2006 and 2013. In 2006, Nangang town and Yandun town in Feixi county were annexed into Shushan district and Baohe district in Hefei city. In 2013, Gaoliu town and Xiaomiao town in Feixi county were annexed into Shushan district in Hefei city. To ensure the comparability of the research data in different time periods, the relevant statistical data for each year is processed based on the latest administrative division in 2016. Given the dramatic adjustment of Feixi administrative division in 2006 and 2013, Nangang town, Yandun town, Gaoliu town, and Xiaomiao town are thus analysed independently to ensure the comparability of the research data. Based on these considerations, this study therefore selects 2002, 2006, 2009, 2012, and 2016 as the time nodes to analyse the spatial process and driving modes of land Urbanization in Feixi county.

## 2.2 Research methodologies

Calculation of Land Urbanization Rate. To date, the calculation of land Urbanization rate is mostly based on the ratio of urban construction land or built-up area to total land area. Land urbanization in counties is different from that in cities since the former has larger non-urban land areas than the latter. If the ratio of urban construction land to total land area is used to measure the land Urbanization rate, the calculated value should be too small to reflect the actual condition of land Urbanization in county areas. Therefore, with reference to existing literature, land Urbanization herein is viewed through the ways in which population settlement is measured. Hence, the proportion of urban construction land in urban and rural construction land is used to measure the land Urbanization rate [37,38].

$$U = \frac{S}{S + L} \times 100\% \tag{1}$$

U is the land Urbanization rate, S is the urban construction land ('urban land' hereafter), including four types of land—urban, construction town, independent industrial and mining land, and urban internal traffic land—and L is rural residential land.

Sector Analysis is used to examine the growth of urban land in different directions in different stages in order to reveal the orientation differentiation of land expansion. To conduct sector analysis in the study area, we select the appropriate centre and radius and divide the study area into several equal fan-shaped areas, and further overlie them with the map layers in different periods to reveal the structural characteristics of construction land in different directions [39]. The main principle of fan-shaped zoning in this study is that the urban land with similar land-use change characteristics in the study area cannot be divided by fan-shaped boundaries as far as possible. Therefore, taking the intersection of national highway G206 and Tangkou road approximately 6 km away from the centre of Feixi county as the centre, and 23.22° east by north as the starting point, the study area is divided into 10 fan-shaped areas with equal angles. By analysing the change of urban land in each fan-shaped area at different stages, the spatial expansion of land Urbanization in Feixi county can be reflected.

Fractal Dimension Index Analysis is used to analyse the fragmentation of urban land. The fractal dimension index can better indicate the fragmentation degree of urban land and the complexity of the irregular boundary of urban land patches. The equation is as follows:

$$PRAC = \frac{2 \ln 0.25P}{\ln A} \tag{2}$$

PRAC is the fractal dimension index, P is the perimeter of the plaque in the study area, and A is the area of the plaque in the study area. Of these, $1 \leq PRAC \leq 2$, the closer PRAC is to 2, the more broken the internal patches of urban land and the more irregular the patch boundary

is. While the closer it is to 1, the stronger the internal filling capacity of urban land, the lower the fragmentation degree and the simpler the patch morphology.

## 3. Spatial expansion characteristics of land urbanization in Feixi county

### 3.1 Change of urban construction land in Feixi county

The overall scale of urban land use in Feixi county increased strikingly from 2002 to 2016. It increased by 7405.17 hm$^2$ in 14 years, from 1938.05 hm$_2$ in 2002 to 9343.22 hm$_2$ in 2016, with an annual growth rate of 11.89% in average (Fig 1). In the meantime, the expansion features of urban land areas vary obviously in different stages. From 2002 to 2009, the urban land area increased from 1938.05 hm$^2$ to 6719.44 hm$^2$, reaching a stunning annual growth rate of 19.44%. After 2009, the growth rate of urban construction land to a great extent slowed down. During 2009–2012, it increased from 6719.44 hm$^2$ to 8087.33 hm$^2$, with an annual growth rate plummeting to 6.37%. In the period of 2012–2016, it continued to expand from 8087.33 hm$^2$ to 9434.22 hm$^2$, with an annual growth rate further decreasing to 3.93%. In short, urban land areas in Feixi county witnessed a rapid and dramatic expansion in the 2000s, but it became more stabilized in scale in the 2010s.

On top of the general change in overall scale in urban land areas in Feixi, the remarkable internal geographical differences are observed (Table 1). It is shown that different towns have distinctive land urbanization trends in both scale and growth rate, which helps us to classify them into different types. The first type refers to towns with large scale and high growth rate, including Taohua, Shangpai, and Zipeng. These towns are the traditional urbanized center in northeastern Feixi, enjoying the greatest proximity to Hefei city proper, specifically Shushan district. The second type refers to towns with small scale and high growth rate, including Huanggang, Fengle, and Yandian. These towns locate in central and southern Feixi, with medium geographical distance to Hefei city proper. The third type refers to towns with small scale and low growth rate, including Sanhe, Guanting, Shannan, Mingchuan, Gaodian, and Shishugang. These towns mostly locate in western Feixi, which is relatively distant to Hefei city proper. The spatial polarization of land urbanization in northeastern, central and southern Feixi stands out, which is sensitive to geographical distance to Hefei city proper. The following

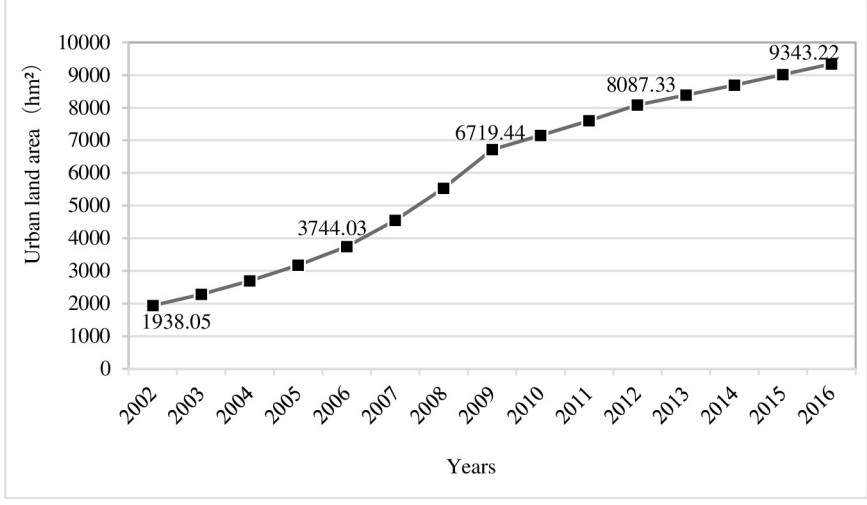

**Fig 1. Changing Scale of urban land areas in Feixi 2002–2016.**

**Table 1. Changes in urban land in towns of Feixi county from 2002 to 2016.**

| Township | 2002 | | 2006 | | | 2009 | | | 2012 | | | 2016 | | | 2002–2016 |
|---|---|---|---|---|---|---|---|---|---|---|---|---|---|---|---|
| | Urban land (hm²) | Proportion (%) | Urban land (hm²) | Proportion (%) | Average annual growth rate (%) | Urban land (hm²) | Proportion (%) | Average annual growth rate (%) | Urban land (hm²) | Proportion (%) | Average annual growth rate (%) | Urban land (hm²) | Proportion (%) | Average annual growth rate (%) | Average annual growth rate (%) |
| Taohua | 550.62 | 28.41 | 1085.1 | 28.98 | 18.48 | 2443 | 36.38 | 31.06 | 2920.7 | 36.13 | 6.13 | 3172.03 | 33.96 | 2.09 | 13.32 |
| Shangpai | 680.74 | 35.12 | 1065.26 | 28.45 | 11.85 | 1531.14 | 22.8 | 12.85 | 2160.07 | 26.72 | 12.15 | 2712.16 | 29.04 | 5.86 | 10.38 |
| Zipeng | 104.04 | 5.37 | 468.47 | 12.51 | 45.67 | 1040.76 | 15.5 | 30.48 | 1148.72 | 14.21 | 3.34 | 1521.65 | 16.29 | 7.28 | 21.12 |
| Huanggang | 111.44 | 5.75 | 219.29 | 5.86 | 18.44 | 474.08 | 7.06 | 29.3 | 523.61 | 6.48 | 3.37 | 576.92 | 6.18 | 2.45 | 12.46 |
| Sanhe | 132.99 | 6.86 | 238.64 | 6.37 | 15.74 | 299.58 | 4.46 | 7.87 | 324.6 | 4.02 | 2.71 | 358.97 | 3.84 | 2.55 | 7.35 |
| Guanting | 106.69 | 5.5 | 208.39 | 5.57 | 18.22 | 317.64 | 4.73 | 15.09 | 344.05 | 4.26 | 2.7 | 349.75 | 3.74 | 0.41 | 8.85 |
| Shannan | 141.61 | 7.31 | 208.4 | 5.57 | 10.14 | 247.1 | 3.68 | 5.84 | 264.64 | 3.27 | 2.31 | 237.27 | 2.54 | -2.69 | 3.76 |
| Fengle | 30.41 | 1.57 | 68.93 | 1.84 | 22.7 | 143.61 | 2.14 | 27.72 | 144.19 | 1.78 | 0.13 | 139.53 | 1.49 | -0.82 | 11.5 |
| Yandian | 17.33 | 0.89 | 54.44 | 1.45 | 33.13 | 63.54 | 0.95 | 5.28 | 81.95 | 1.01 | 8.85 | 110.86 | 1.19 | 7.84 | 14.17 |
| Mingchuan | 23.53 | 1.21 | 60.2 | 1.61 | 26.48 | 84.98 | 1.27 | 12.18 | 83.48 | 1.03 | -0.59 | 87.82 | 0.94 | 1.27 | 9.86 |
| Gaodian | 15.12 | 0.78 | 25.99 | 0.69 | 14.5 | 28.43 | 0.42 | 3.04 | 44.96 | 0.56 | 16.5 | 39.28 | 0.42 | -3.32 | 7.06 |
| Shishugang | 23.53 | 1.21 | 40.92 | 1.09 | 14.84 | 42.06 | 0.63 | 0.92 | 42.83 | 0.53 | 0.61 | 33.47 | 0.36 | -5.98 | 2.55 |

**Data sources:** USGS remote sensing interpretation data (2002), land use change survey data of Feixi county Bureau of Land and Resources (2006–2016).

subsection will offer a more detailed investigation of the spatial expansion of land urbanization in Feixi.

## 3.2 Evolution of urban land expansion directions in Feixi county

According to the findings derived from the sector analysis, from 2002 to 2016, the second, third, and tenth quadrants had faster urban land expansion speed, exceeding around 15% of the average annual growth rate. Accordingly, the quadrants with a larger scale of urban land growth comprise the first, second, third, fourth, and tenth quadrants, with a land use growth of more than 500 hm$^2$. Most of the faster-growing quadrants are distributed in the North, East, and Northeast of Feixi county, in other words, the areas neighboring the Southwesten city proper in Hefei. Interestingly, in the post-2009 stage, along with the slowing growth rate in general, the spatial restrutcuirng in land urbanization is as well occurring. During 2012–2016, the scale of urban land in the third and tenth quadrants instead of the first and second quadrants expanded more significantly, largely due to the saturation of traditional urbanized center in northeastern Feixi county near Hefei city proper and its spillover effects. Moreover, it should be noted that the expansion of urban land in the sixth quadrant has been increasingly rapid, which is mainly benefitted from its geographical location along the main economic connection direction of Feixi county linked by the main traffic line. In the fifth and eighth quadrants, where Shannan town and Sanhe town are relatively away from the main urbanized area as well as the main economic connection directions, the urban land expansion during 2002–2016 remains at a low level.

## 3.3 Evolution of urban land use spatial pattern in Feixi county

From 2002 to 2016, the spatial structure of urban land areas in Feixi county experienced dramatic expansion, which changed from highly fragmented and weakly linked to strongly linked. In 2002, the distribution of urban land there was highly fragmented and weakly linked. The traditional urbanized center, the old area of Taohua Industrial Park neighboring Hefei city proper, and the surrounding main traffic lines operated as the only hot spot. From 2002 to 2006, the urban land use structure of Feixi county was moving towards an emerging polycentric structure. Apart from the old center in Shangpai town, the rapid expansion of urban land in Taohua town and Zipeng town was the main contributor. The rapid growth of urban land in Feixi county during 2006–2009 played a key role in shaping the polycentric spatial structure in Feixi county, with a tripolar structure underpinned by Shangpai town, Taohua town, and Zipeng town.

In the post-2009 period, however, the spatial restructuring of urban land use in Feixi county has been much deviated from the pre-2009 pattern. First, the spatial expansion slowed down to a great extent, which further stabilized rather than enhanced the polycentric spatial structure. Second, in addition to the leading three towns in land urbanization, other towns along major traffic corridors began to embrace faster and more dramatic urban land expansion, such as Huanggang, Fengle, and Yandian. Dissimilar with the contiguous expansion towards all directions in towns neighboring Hefei city proper before 2009, the urban land use expansion in post-2009 became more spatially selective and traffic line oriented.

## 3.4 Evolution of urban land use spatial form in Feixi county

The Fractal Dimension Index is further deployed to reflect the complexity of urban land use form. The evolution process of the urban land use spatial form in Feixi county was analysed in conjunction with the growth mode of landscape pattern. Considering the realities of urban land use change in Feixi county, FRAGSTATS 4.2 software is used to calculate eight

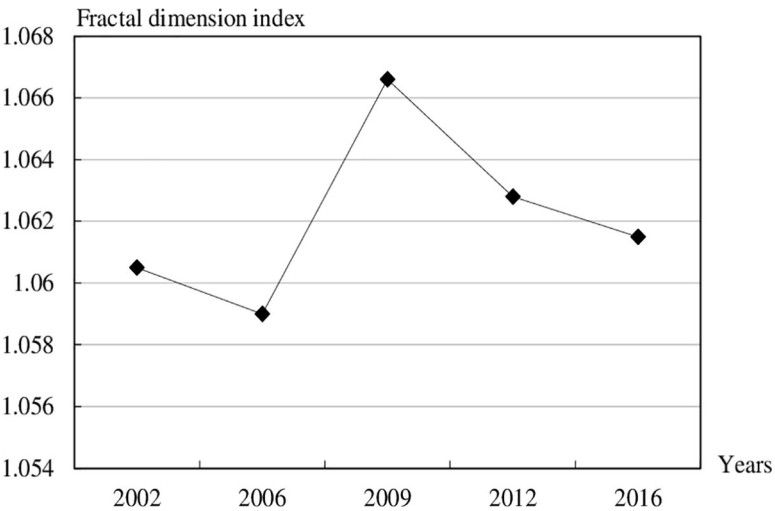

**Fig 2. Fractal dimension index of urban land use form in Feixi county from 2002 to 2016.**

neighbourhood fractal dimension index methods to analyse the fragmentation degree of urban land in Feixi county. The results show that the fractal dimension index of urban land use form in Feixi county from 2002 to 2016 is fluctuating around 1.062 (Fig 2). It witnessed an rapid increase before 2009 and a remarkable plummet since 2009. Accordingly, the spatial distribution pattern of urban land is relatively simple.

From 2002 to 2006, the fractal dimension index decreased by 0.0015, with an insignificant change. During this period, the growth pattern of urban land was more diverse. The different ways of obtaining urban land data in 2002 and 2006 may also had a certain impact on this. From 2006 to 2009, the fractal dimension index increased by 0.0051, and the degree of irregularity of urban land correspondingly increased. During this period, the urban land in Feixi county expanded rapidly. In addition to the marginal growth of the main urban land in the traditionally urbanized town areas, there was more urban land characterized by an enclave growth pattern. As a result, the fractal dimension index of land use increased and urban land was more fragmented. From 2009 to 2016, the fractal dimension index in turn began to decrease. In the meantime, there were two growth modes of urban land use: marginal growth and infill growth. With the continuous growth of urban land in Feixi, urban land gradually tended to be saturated. Feixi county has higher requirements on land use and pays more attention to efficient and intensive use. Regular marginal growth and internal filling growth became the main modes of urban land growth in this period.

In addition, due to the demands for urban development and expansion of Hefei city, Feixi county successively allocated the whole construction system of Nangang town, Yandun town, Xiaomao town, and Gaoliu town to the jurisdiction of Hefei city proper in 2006 and 2013. From 2002 to 2006, the urban land of Nangang town increased from 232.85 hm$^2$ to 576.01 hm$^2$, with an increase of 343.16 hm$^2$; the urban land of Yandun township increased from 69.85 hm$^2$ to 411.64 hm$^2$, with an increase of 341.79 hm$^2$. The urban land use in Nangang town and Yandun town mainly increased by filling type, and the urban land is mainly distributed at the boundary of Feixi county and Hefei city proper and the surrounding areas along the traffic lines. In 2002, the urban land use of Xiaomiao town and Gaoliu town was 193.23 hm$^2$ and 114.90 hm$^2$, respectively. By 2006, the urban land use of Xiaomiao town and Gaoliu town had increased to 356.13 hm$^2$ and 166.67 hm$^2$, respectively. The urban land use of Xiaomiao town grew rapidly, mainly in the form of enclave growth, but the spatial distribution of urban land

was relatively scattered. While the urban land use of Gaoliu town grew relatively slowly. As Nangang town was placed under the jurisdiction of Shushan district of Hefei city and became part of Hefei city proper in 2009, the urban land of Xiaomao town increased to 510.52 hm$^2$, primarily in the form of marginal growth. The urban land distribution became increasingly compact, and the expanded land was mainly distributed in the side of Xiaomao town near Nangang town. Furthermore, due to the construction of Hefei Xinqiao National Airport, the urban land use in Gaoliu town witnessed a rapid growth as well. From 2006 to 2009, the urban land use increased by 600.68 hm$^2$. However, after the completion of these infrastructure projects, the growth of urban land in Xiaomiao town and Gaoliu town slowed down in the post-2009 period. The urban land in Xiaomiao town only increased by 42.45 hm$^2$, while the urban land in Gaoliu town only increased by 96.08 hm$^2$.

## 4 Driving forces of land urbanization in Feixi County

The rapid growth of urban land in Feixi county in the 2000s is mainly attributable to the accelerated urbanization process in China and the remarkable land-centred expansion of its 'strong' neighbour, Hefei city as the provincial capital in Anhui province in Central China. Moreover, the slow-down of its land expansion in the post-2009 period is highly associated with a new wave of Chinese urban transformation towards new-type urbanization and city-regionalization. The particular characters of its urban land expansion are embedded in its specific regional context (e.g., neighboring to Hefei city proper specifically Shushan district, functioning as a key node in the Yangtze River Delta city-region). Based on both statistical and qualitative evidence, According to the degree of government participation, three driving forces of land urbanization in Feixi county since the early 2000s are summarized as follows, government and market participation are roughly equal in central city expansion, the government plays an absolute leading role in government-led park construction, the market plays an absolute leading role in large industrial projects along traffic corridors.

### 4.1 Central city expansion

Through the analysis of the urban land use scale of Hefei city proper and its adjacent counties of Feixi county, Feidong county, and Changfeng county in different periods, this study discusses the relevance of urban land change between the central city and its counties, and further explains the relationship between urban land use change between Feixi county and Hefei city proper. The results show that, in addition to this stage in 1995–2002, the average annual growth rate of urban land in each district and county was below 5%. In the rest of the four stages of each district and county urban area, the average annual growth rate was over 5%, and the average annual growth rate of urban land in each county exceeded that of Hefei city proper. However, from the perspective of absolute growth scale, the increment of urban land in each stage of Hefei is more than twice that of each county area. In addition, compared with other districts and counties, Feixi county had the fastest growth rate of urban land use from 2006 to 2009, with an average annual growth rate of 15.75%. During this period, the development centre of Hefei city moved to the south, the development of Hefei High-tech Industrial Development Zone was accelerated, and urban land in Binhu New Area of Hefei city expanded rapidly, which largely led to the rapid growth of urban land in Feixi county. From 2012 to 2016, the gap of annual growth rate of urban land among each district and county began to narrow. The growth of urban land in Hefei was mainly characterized as internal filling growth, while the growth of urban land in Feixi, Feidong, and Changfeng counties showed increasingly obvious regional agglomeration.

Generally speaking, the expansion of urban land in counties around Hefei city proper is similar. The areas with rapid growth of urban land are distributed in the areas adjacent to Hefei city proper, which are greatly affected by Hefei urban expansion. For example, the main urban area of Changfeng county, which is located in Shuihu town far away from the main urban area of Hefei city, is in the north of the county area. By 2016, Shuangdun town and Gangji town, located in the north of Hefei city, had a great expansion in urban land use. At the same time, however, the possibility of urban land supply and the accumulation of industry in each county also affected the expansion direction of urban land in Hefei to a certain extent. The above analysis shows that the analysis of the mode of land Urbanization in Feixi county cannot ignore the background of Feixi county as a suburban county of Hefei city. The development of Feixi county is closely related to the expansion Hefei city proper in both intensity and direction. The rapid expansion of urban land in Feixi county is related to the demand of its own development and the radiation and land demand of Hefei city. The land Urbanization of Feixi county is carried out under the guidance of Hefei city and taking into account its own development.

In addition, according to the interpretation of results of remote sensing images, Nangang town and Yandun township have become one of the main areas of land expansion in Hefei city in 2006, and the scale of urban land has increased rapidly. Since Xiaomiao town and Gaoliu town were included in Hefei city administratively in 2013, the expansion of urban land has been relatively slow, because it is mainly used as preparatory land for urban spatial expansion of Hefei city proper. Due to the influence of administrative division adjustment, after some towns of Feixi county were included in Hefei City proper, the urban land expanded rapidly and the land use efficiency improved greatly, which can also be regarded as an indirect form of land Urbanization in Feixi county.

## 4.2 Government-led park construction

In the 1990s, China entered the climax of development zone construction, with development zones, industrial parks, and township industrial parks springing up all over the country (Cartier, 1990). Taohua Industrial Park in Feixi county was also established in this period.Since then, Feixi relied on the construction of Taohua Industrial Park to raise its efforts to attract investment, thereby prompting a number of well-known large-scale enterprises to enter the park for construction. The demonstration and driving role of Taohua Industrial Park was gradually highlighted. In 2002, Taohua Industrial Park New Area (located in Yandun township, close to Hefei Economic and Technological Development Zone) began its planning and construction. In 2005, Feixi county and Hefei High-tech Industrial Development Zone cooperated to establish Baiyan Science and Technology Park with a development area of approximately 7.7 square kilometres in accordance with the principle of 'unchanged zoning, unchanged household registration, cooperative development, benefit sharing and overall development'. In the same year, Feixi county government proposed speeding up the construction of 'two parks and eight township industrial concentration areas'. Subsequently, Zipeng town industrial cluster area, Sanhe town industrial cluster area, Shangpai town industrial cluster area, and Xiaomiao town industrial cluster area were approved and developed. During this period, Feixi county put forward the goal of speeding up the development of township industry by means of revitalising stock and land replacement, so as to further improve the agglomeration and scale effect of township industry and improve the efficiency of land use. By 2006, Taohua Industrial Park had grown from 2.97 square kilometres to 8.1 square kilometres. In the same year, Taohua Industrial Park proposed cooperating with Taohua town to build the industrial park, and took the urban construction land south of Hefei Wuhan Railway as the

expansion area of Taohua Industrial Park, further underpinning the expansion of the Taohua Industrial Park. In 2008, Feixi county and Hefei Economic and Technological Development Zone jointly developed Xingang Industrial Park based on the principle of 'unchanged zoning, complementary advantages, cooperative development, and benefit sharing'. In 2009, the cooperation agreement was signed on the second phase of Xingang industrial park with an area of 56 square kilometres. In the same year, the planning and construction of 10 square kilometres of Baiyan Industrial Park Phase II began. In 2015, the construction of Xingang south district commenced [Feixi county *'government work report'* (2005, 2008), *'Feixi county Industrial Enterprise Yearbook 2011'*, and so on.].

The government of Feixi county vigorously promoted the construction of industrial parks and industrial agglomeration areas in towns and townships, which plays a leading role in the growth of urban land use in Feixi county. In addition to the relatively rapid expansion of urban land in some towns along the main traffic lines, the growth of urban land in Feixi county is mainly located in and around the industrial (agglomeration) park area. That is, the park construction under the government's guidance promotes the rapid growth of urban land use in Feixi. At the same time, the rapid development of the park has a significant driving effect on its surrounding areas. On this basis, a certain spill-over effect has been produced, which makes Feixi county pay more attention to the efficient use of urban land. It then entered the stage of land stock development and structural adjustment.

## 4.3 Large industrial projects along traffic corridors

Before 2002, at the initial stage of county industrialisation, the number of industrial enterprises in Feixi county was relatively small. In 2001, there were only 23 industrial enterprises above the designated scale in Feixi county, with an added value of 172 million yuan. Moreover, the area occupied was relatively small and the agglomeration effects was immature. There was no large-scale expansion of urban land. Through the interpretation of remote sensing images in the corresponding years, 1995–2001, except for the expansion of Peach Blossom Industrial Park in Feixi county, the overall expansion of urban land in Feixi county was relatively small, mainly based on the marginal growth of key towns. By 2001, the Taohua Industrial Park in Feixi county covered an area of 2.97 square kilometres, and had formed a certain industrial cluster, breaking through the annual output value of 1 billion yuan. The construction of Anhui Anli Material Technology Co., Ltd., Anhui Jianghuai Automobile Co., Ltd., and Anhui Hongyuan Electric Tower Manufacturing Co., Ltd., and other large-scale well-known enterprises entered the park to promote the later development of Feixi county and create a good industrial milieu. However, in general, before 2002, the industrial output value of Feixi county was relatively low and the scale of enterprises was small. By 2002, the industrial added value only accounted for 19.66% of the GDP of that year, and the industrial added value above the designated size only accounted for 37.66% of the industrial added value (see Table 2). With the low-speed development of industry, the urban land use of Feixi county was relatively slow before 2002.

Since 2002, the industrial output value of Feixi county has been growing rapidly. The proportion of the added value of Industrial Enterprises above the designated scale in the total industrial added value has increased rapidly to more than 70%, and the scale effect of industries and Industrial Enterprises above the designated scale has gradually strengthened. Particularly after 2005, Feixi county government seized the strategic opportunity period of 'large demolition, large construction' and scientific and technological innovation in pilot city construction, highlighted the east development strategy, innovated the land development mode, vigorously promoted land replacement, and guided the rational layout of key industries,

**Table 2. Industrial enterprise output value statistics from Feixi county.**

| years | GDP (bn yuan) | Industry (bn yuan) | Ratio of industry to GDP (%) | Added value of industries above designated size (bn yuan) | Ratio of industries above designated size to industry (%) | Number of industries above designated size |
|---|---|---|---|---|---|---|
| 2000 | 2.04 | 4.01 | 19.66 | 1.51 | 37.66 | - |
| 2001 | 2.204 | 5.81 | 26 | 1.72 | 39.08 | - |
| 2002 | 2.64 | 6.96 | 26.36 | 5.04 | 72.41 | - |
| 2003 | 3.61 | 15 | 41.55 | - | - | - |
| 2004 | 5.481 | 18.68 | 34.08 | 15.81 | 84.64 | - |
| 2005 | 6.905 | 24.22 | 35.08 | 20.84 | 86.04 | 58 |
| 2006 | 9.55 | 43.4 | 45.45 | 35.7 | 82.26 | 89 |
| 2007 | 12.02 | 54.9 | 45.67 | 46.2 | 84.15 | 137 |
| 2008 | 15.36 | 72.7 | 47.33 | 65.5 | 90.1 | 200 |
| 2009 | 21.46 | 107.6 | 50.14 | 99.2 | 92.19 | 287 |
| 2010 | 27.48 | 145.5 | 52.95 | 141.32 | 97.13 | - |
| 2011 | 33.437 | 190.2 | 56.88 | 181 | 95.16 | 286 |
| 2012 | 41.67 | 235.3 | 56.47 | 204.5 | 86.91 | 334 |
| 2013 | 46.4 | 267.4 | 57.63 | 234.6 | 87.73 | 389 |
| 2014 | 50.88 | 312 | 61.32 | 277.6 | 88.97 | 409 |
| 2015 | 55.185 | 338.37 | 61.32 | 277.7 | 82.07 | 423 |
| 2016 | 60.502 | 356.58 | 58.94 | 225.52 | 63.25 | 430 |

**Note:** The data in this table are based on the statistical bulletin and statistical yearbook data of Feixi county in the corresponding year, and '-' represents abnormal or non-statistical data.

ushering in the rapid expansion of urban land in Feixi county. With the continuous improvement of the scale of industrial enterprises, Amway materials, Jianghuai Automobile, Gree Electric appliances, and other large-scale enterprises have occupied a large area, and play a strong driving role. The spill-over effect is constantly emerging, which makes the industrial production space in the park more centralised. At the same time, Feixi county focuses on key industrial enterprises to vigorously attract investment for its upstream and downstream industries, and is gradually achieving the transformation from scattered investment promotion to industrial chain and industrial cluster investment. It also provides greater policy stimulation on land demand for key projects in line with its development orientation, which promotes the rapid growth of urban land in Feixi county, while the land use growth shows agglomeration and contiguous pattern growth.

Particularly after 2009, along with the city-regionalization progress across China, the spill-over effect of Feixi to large-scale industrial projects began to appear especially along key traffic corridors in the Yangtze River Delta city-region. Feixi county began to pay attention to the connotative development and improvement of county comprehensive functions on the basis of the early development, and the infrastructure construction such as road and railway traffic made great progress. In 2010, Paihe Avenue, Jinzhai South road, and other regional commercial plates appeared along the main road. The establishment of Huanan city comprehensive commerce and trade logistics centre in the southwest of Taohua Town, the approval of Hefei export processing zone of Xingang Industrial Park, and the planning and construction of Port Logistics Industrial Zone in the Southeast of Feixi main urban area were the main driving forces of urban land increase in Feixi county during this period. In 2013, the '1331 strategy' of Hefei city proposed Feixi as the southwest area of the main city of Hefei, prompting Feixi

county to pay more attention to the functional integration and industrial transformation and upgrading of the county. To speed up the connection with the main urban area of Hefei, Feixi needs more land for infrastructure such as roads and occupies a large area of urban land. The key projects brought by investment promotion in Feixi county began to focus more on quality than quantity, and the demand for industrial land was relatively slow [according to field research and interviews]. In this period, the integration of industry and city, business logistics and the tertiary industry were further developed.

## 5 Conclusion and discussion

The remarkable urbanization in post-reform China has great contribution to the advent of 'Global Urbanism', and this process continues to prevail since the 2010s. However, it should be noted that the geographical locus of urbanization in China has been shifting from highly urbanized eastern coastal areas to relatively underdeveloped central and western regions under the background of new-type urbanization strategy. In particular, county-level urbanization is much emphasized to facilitate urban-rural integration during this new wave of urban transformation. As it is widely known that Chinese urbanization is characterized as 'land-centred', land urbanization process and driving forces in counties in Central and Western China have become increasingly intriguing.

This paper therefore examines the land urbanization process and driving forces of Feixi county in Anhui province in Central China. Neighboring Hefei city proper, Feixi county is equipped with abundant as well as geographically sensitive urbanization dynamics, which represents an appropriate case for us to better understand the urbanization process in Central China that is somehow dissimilar to the rich evidence in coastal regions.

As for spatial expansion characters, the overall scale of urban land in Feixi county increased remarkably during 2002–2016. It expanded from 1938.05 hm$^2$ in 2002 to 9343.22 hm$^2$ in 2016, with an average annual growth rate of 11.89%. However, the temporal variation is obvious. Before 2009, the land expansion in Feixi county is highly stunning, reaching an annual growth rate of 19.44%. In stark contrast, in the post-2009 period, the annual growth rate decreased to 6.37%. The pre-2009 period witnessed extensive marginal growth, while the post-2009 period switched towards internal filling growth. Thus, it is mirrored that the year 2009 is a cutting line between quantity-based urbanization and quality-based urbanization in Feixi county.

Geographically, during the pre-2009 period, the agglomeration trend of urban land scale is strengthening, particularly in towns close to the urban area of Hefei and towns around Chengguan town of Feixi county. In the post-2009 period, the polycentric spatial structure of Feixi county was further strengthened, and the urban land structure began to be stable. By 2016, the multi-centre area with Shangpai town and Taohua town as the main component of urban land showed a certain network and group structure due to the improvement of road traffic and other basic implementation, as well as their proximity. In addition, the growth of urban land in Zipeng town shows a block and group expansion, and the growth of urban land in some regions shows a zonal development trend along the traffic line. This research finding further verifies Chen's [40] research conclusion.

Driving forces of land urbanization in Feixi county is summarized as central city expansion, government-led industrial park construction, and large industrial projects along traffic corridors. First, affected by the polarisation effect and spillover effect of Hefei city proper, the expansion of urban land in Feixi county is closely related to the growth of urban land in Hefei city as a whole. In the context of urban-rural integration and city leading county, the intensity, distribution, and velocity of urban land expansion in Feixi county cannot be isolated from Hefei city. Second, the establishment of industrial parks including city-county cooperation

parks under the scale expansion of Hefei High-tech Industrial Development Zone contributes dramatically to the land urbanization in Feixi county. Government-led park construction offers spatial platforms for county industrialization and land urbanization. Third, large industrial projects along key traffic corridors in the context of city-regionalization enrich land urbanization dynamics in Feixi county. In addition to marginal and internal filling growth, geographically selective industrialization and urbanization along traffic corridors become fresh momentum for land urbanization in Feixi county, which is underpinned by the functional division within a wider spatial order, for example, city-regions or urban clusters.

In this study, remote-sensing data, the first national land change survey data, and the second national land change survey data were combined to ensure the continuity of the study in the time period. However, the lack of comparability between different data may have had a certain impact on the research results. In the future, with the further refinement of land survey data and the further improvement of data acquisition technology, the research data of land urbanization in county areas will become richer and more accurate. Therefore, the comparative analysis of different types of urban land use changes will likely provide a fruitful direction of land Urbanization research in county areas in the future. From the perspective of better realizing the goal of land urbanization, according to the conclusions of this study, first, it is suggested to scientifically formulate and strictly implement the land use planning and strengthen the leading role of the planning in land urbanization; Second, for similar areas in central China, land supply should be effectively matched according to different development stages to improve land use efficiency.

## Supporting information

**S1 Data.**
(DOCX)

## Author Contributions

**Conceptualization:** Huafu Jiao, Xianzhong Cao.

**Data curation:** Xiangxiang Zhang.

**Formal analysis:** Xiangxiang Zhang, Chao Yang.

**Funding acquisition:** Huafu Jiao.

**Investigation:** Chao Yang.

**Methodology:** Xiangxiang Zhang, Chao Yang.

**Software:** Chao Yang.

**Visualization:** Xiangxiang Zhang, Chao Yang.

**Writing – original draft:** Xiangxiang Zhang, Chao Yang.

**Writing – review & editing:** Huafu Jiao, Xianzhong Cao.

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
