## [Decision Letter · Decision Letter 0]

7 Jan 2021

PONE-D-20-39602

The Characteristics of Spatial Expansion and Driving Forces of Land Urbanisation in Counties in Central China: A Case Study of Feixi County in Hefei City

PLOS ONE

Dear Dr. Cao,

Thank you for submitting your manuscript to PLOS ONE. After careful consideration, we feel that it has merit but does not fully meet PLOS ONE’s publication criteria as it currently stands. Therefore, we invite you to submit a revised version of the manuscript that addresses the points raised during the review process.

We look forward to receiving your revised manuscript.

Kind regards,

Mingxing Chen, Ph.D.

Academic Editor

PLOS ONE

Journal Requirements:

2.We note that Figure(s) 1, 3, 4, 6 and 7 in your submission contain map images which may be copyrighted. All PLOS content is published under the Creative Commons Attribution License (CC BY 4.0), which means that the manuscript, images, and Supporting Information files will be freely available online, and any third party is permitted to access, download, copy, distribute, and use these materials in any way, even commercially, with proper attribution. For these reasons, we cannot publish previously copyrighted maps or satellite images created using proprietary data, such as Google software (Google Maps, Street View, and Earth). For more information, see our copyright guidelines: http://journals.plos.org/plosone/s/licenses-and-copyright.

a.    You may seek permission from the original copyright holder of Figure(s) 1, 3, 4, 6 and 7 to publish the content specifically under the CC BY 4.0 license. 

3. Please amend your authorship list in your manuscript file to include author Xiangxiang Zhang.

Please also include links to where the datasets can be accessed 'including the Landsat5 and Statistical Yearbook data used in the study.

5. Please ensure that you include a title page within your main document. We do appreciate that you have a title page document uploaded as a separate file, however, as per our author guidelines (http://journals.plos.org/plosone/s/submission-guidelines#loc-title-page) we do require this to be part of the manuscript file itself and not uploaded separately.

Reviewers' comments:

Reviewer's Responses to Questions

**Comments to the Author**

1. Is the manuscript technically sound, and do the data support the conclusions?

Reviewer #1: Yes

Reviewer #2: Yes

2. Has the statistical analysis been performed appropriately and rigorously? 

Reviewer #1: Yes

Reviewer #2: Yes

3. Have the authors made all data underlying the findings in their manuscript fully available?

Reviewer #1: No

Reviewer #2: Yes

4. Is the manuscript presented in an intelligible fashion and written in standard English?

Reviewer #1: Yes

Reviewer #2: Yes

5. Review Comments to the Author

Reviewer #1: This paper works on an interesting topic about urbanization, taking the Feixi county in Anhui province in Central China as a case study. Under the support of both quantitative and qualitative methodologies, this paper seeks to investigate the spatial expansion and driving forces of land urbanization, it aims to unveil emerging land urbanization realities in the specific context of central China. The referee would suggest a minor revision.

First, literature review needs to be strengthened, especially the past literature on the evaluation of land urbanization. According to the literature review, it is suggested to further clarify the research problem.

Second, according to the results, it is necessary to compare with the existing research on land urbanization and find out the unique contribution of this research. Meanwhile, as key words, the urban-rural integration, city regionalization were not fully combined with the research questions and reflected in conclusion and discussion.

Third, the results of the study are not accurately expressed, such as “From 2002 to 2016, the spatial structure of urban land areas in Feixi county experienced dramatic expansion.” While the author(s) failed to interpret it deeply.

Forth, there is a clerical error in sentence: “particular, county-level urbanization is much emphasized to facilitate urban-rural integration during this new wave of urban transformation ()”. What’s the “()” mean here?

Reviewer #2: I am pleased to read your manuscript. The manuscript mainly analysis the spatial expansion characteristics and the driving forces of land urbanization in Feixi county. Some issues are suggested to be addressed.

1. The theoretical basis of the manuscript is not solid enough, especially in the driving force of land urbanization expansion, so it is difficult to reflect the academic significance and innovation of the manuscript. I recommend making effort to find an appropriate and adequate theoretical background.

2. The main body of the manuscript is about the spatial expansion and driving force of land urbanization, however, the “new-type urbanization”, “urban-rural integration” of the Keyword aren’t related about the main issue of the manuscript.

3. Figure 1. Area map of Feixi county. and the related content should be recognized in the part 2 Materials and Methods.

4. In the abstract, it is mentioned that its process and driving forces in counties beyond the Eastern coastal areas are less known. The driving forces in counties in central China is the important innovation in the manuscript, so I suggest the author add the literature comparison of driving force of land urbanization in different regions in the part 4 or part 5, so as to highlight the literature contribution of the manuscript.

5. In part 4: Driving Forces of Land Urbanization in Feixi County, the manuscript mainly mention 3 kind driving forces of land urbanization: central city expansion, government-led industrial park construction, and large industrial projects along traffic corridors. However, the connection between the three driving forces is lack.

6. There are some slight mistakes in the manuscript. For example, the format of references, punctuation in the table should be unified:

Page 4：Fractal Dimension Index Analysis is used to analysis the fragmentation of urban land. The fractal dimension index can better indicate the fragmentation degree of urban land and the complexity of the irregular boundary of urban land patches [26].

Page 9：The evolution process of the urban land use spatial form in Feixi county was analyzed in conjunction with the growth mode of landscape pattern [27].

Page 11：The rapid expansion of urban land in Feixi county is related to the demand of its own development and the radiation and land demand of Hefei city. The land urbanization of Feixi county is carried out under the guidance of Hefei city and taking into account its own development.

Page 12：In the 1990s, China entered the climax of development zone construction, with development zones, industrial parks, and township industrial parks springing up all over the country [29](Cartier, 1990).

Page 15：In particular, county-level urbanization is much emphasized to facilitate urban-rural integration during this new wave of urban transformation ().

Page7: Table 1: Urban land (hm2).

7. The spelling of this manuscript requires proofreading and I recommend English language editing by a native speaker.

6. PLOS authors have the option to publish the peer review history of their article (what does this mean?). If published, this will include your full peer review and any attached files.

Reviewer #1: No

Reviewer #2: No

---

## [Author Response · Author response to Decision Letter 0]

1 Mar 2021

Dear,

Thank you very much for the comments of the reviewers. We have carefully revised them. The following is our response:

Reviewer #1: This paper works on an interesting topic about urbanization, taking the Feixi county in Anhui province in Central China as a case study. Under the support of both quantitative and qualitative methodologies, this paper seeks to investigate the spatial expansion and driving forces of land urbanization, it aims to unveil emerging land urbanization realities in the specific context of central China. The referee would suggest a minor revision.

First, literature review needs to be strengthened, especially the past literature on the evaluation of land urbanization. According to the literature review, it is suggested to further clarify the research problem.

Reply: Thanks very much. We have strengthened the literature review, such as Land urbanization was first proposed by Lu (2007) in 2006, it means that in a certain country and region, due to the advancement of urbanization, land has been gradually changed to urban…, 

The United Nations proposed in the 2030 Agenda for Sustainable Development Goals (SDGs), SDG11.3.1 has been used to evaluate land urbanization(United Nations General Assembly, 2015)…

The remainder of this paper is organized as follows. The next section introduces the materials and methods. Section 3 describes the spatial expansion characteristics of land urbanization in Feixi county. Section 4 analyses the driving forces of land urbanization in Feixi county. The last section concludes the paper.

Second, according to the results, it is necessary to compare with the existing research on land urbanization and find out the unique contribution of this research. Meanwhile, as key words, the urban-rural integration, city regionalization were not fully combined with the research questions and reflected in conclusion and discussion.

Reply: Thanks very much. We have compare with the existing research on land urbanization, and find that central city expansion, government-led industrial park construction, and large industrial projects along traffic corridors has an important impact on land urbanization. Such as Particularly after 2009, along with the city-regionalization progress across China, the spill-over effect of Feixi to large-scale industrial projects began to appear especially along key traffic corridors in the Yangtze River Delta city-region (Lin, 2014; Wu, 2016)…

Geographically, during the pre-2009 period, the agglomeration trend of urban land scale is strengthening, particularly in towns close to the urban area of Hefei and towns around Chengguan town of Feixi county. In the post-2009 period, the polycentric spatial structure of Feixi county was further strengthened, and the urban land structure began to be stable. By 2016, the multi-centre area with Shangpai town and Taohua town as the main component of urban land showed a certain network and group structure due to the improvement of road traffic and other basic implementation, as well as their proximity. In addition, the growth of urban land in Zipeng town shows a block and group expansion, and the growth of urban land in some regions shows a zonal development trend along the traffic line. This research finding further verifies Chen's (2014) research conclusion.

Key words have been revised, “land Urbanization; spatial Expansion; driving Forces; Feixi county”.

Third, the results of the study are not accurately expressed, such as “From 2002 to 2016, the spatial structure of urban land areas in Feixi county experienced dramatic expansion.” While the author(s) failed to interpret it deeply.

Reply: Thanks very much. We have revised the sentence,“From 2002 to 2016, the spatial structure of urban land areas in Feixi county experienced dramatic expansion, which changed from highly fragmented and weakly linked to strongly linked”.

Forth, there is a clerical error in sentence: “particular, county-level urbanization is much emphasized to facilitate urban-rural integration during this new wave of urban transformation ()”. What’s the “()” mean here?

Reply: Thanks very much. () is a spelling mistake, We have deleted it.

Reviewer #2: I am pleased to read your manuscript. The manuscript mainly analysis the spatial expansion characteristics and the driving forces of land urbanization in Feixi county. Some issues are suggested to be addressed.

1. The theoretical basis of the manuscript is not solid enough, especially in the driving force of land urbanization expansion, so it is difficult to reflect the academic significance and innovation of the manuscript. I recommend making effort to find an appropriate and adequate theoretical background.

Reply: Thanks very much. The era of ‘Global Urbanism’ has been dawning since the onset of the 21st century, as a consistently increasing number of the world’s population settle in urban areas. In the meantime, the world urbanization process is witnessing ‘new geographies’ which is characterized as the dislocation of focal urbanizing territories from the Global North to Global South. There has been a problem of uncoordinated population and land urbanization. Land urbanization was first proposed by Lu (2007) in 2006, it means that in a certain country and region, due to the advancement of urbanization, land has been gradually changed to urban. 

2. The main body of the manuscript is about the spatial expansion and driving force of land urbanization, however, the “new-type urbanization”, “urban-rural integration” of the Keyword aren’t related about the main issue of the manuscript.

Reply: Thanks very much. We did not discuss the “new-type urbanization”, “urban-rural integration” in the article, So the key words have been revised, “land Urbanization; spatial Expansion; driving Forces; Feixi county”.

3. Figure 1. Area map of Feixi county. and the related content should be recognized in the part 2 Materials and Methods.

Reply: Thanks very much. We have moved the Figure 1 to the Figure 1 part 2 Materials and Methods.

4. In the abstract, it is mentioned that its process and driving forces in counties beyond the Eastern coastal areas are less known. The driving forces in counties in central China is the important innovation in the manuscript, so I suggest the author add the literature comparison of driving force of land urbanization in different regions in the part 4 or part 5, so as to highlight the literature contribution of the manuscript.

Reply: Thanks very much. We have compare with the existing research on land urbanization, and find that central city expansion, government-led industrial park construction, and large industrial projects along traffic corridors has an important impact on land urbanization. Such as Particularly after 2009, along with the city-regionalization progress across China, the spill-over effect of Feixi to large-scale industrial projects began to appear especially along key traffic corridors in the Yangtze River Delta city-region (Lin, 2014; Wu, 2016)…

Geographically, during the pre-2009 period, the agglomeration trend of urban land scale is strengthening, particularly in towns close to the urban area of Hefei and towns around Chengguan town of Feixi county. In the post-2009 period, the polycentric spatial structure of Feixi county was further strengthened, and the urban land structure began to be stable. By 2016, the multi-centre area with Shangpai town and Taohua town as the main component of urban land showed a certain network and group structure due to the improvement of road traffic and other basic implementation, as well as their proximity. In addition, the growth of urban land in Zipeng town shows a block and group expansion, and the growth of urban land in some regions shows a zonal development trend along the traffic line. This research finding further verifies Chen's (2014) research conclusion.

5. In part 4: Driving Forces of Land Urbanization in Feixi County, the manuscript mainly mention 3 kind driving forces of land urbanization: central city expansion, government-led industrial park construction, and large industrial projects along traffic corridors. However, the connection between the three driving forces is lack.

Reply: Thanks very much. The rapid growth of urban land in Feixi county in the 2000s is mainly attributable to the accelerated urbanization process in China and the remarkable land-centred expansion of its ‘strong’ neighbour, Hefei city as the provincial capital in Anhui province in Central China. Moreover, the slow-down of its land expansion in the post-2009 period is highly associated with a new wave of Chinese urban transformation towards new-type urbanization and city-regionalization. The particular characters of its urban land expansion are embedded in its specific regional context (e.g., neighboring to Hefei city proper specifically Shushan district, functioning as a key node in the Yangtze River Delta city-region). Based on both statistical and qualitative evidence, According to the degree of government participation, three driving forces of land urbanization in Feixi county since the early 2000s are summarized as follows, government and market participation are roughly equal in central city expansion, the government plays an absolute leading role in government-led park construction, the market plays an absolute leading role in large industrial projects along traffic corridors.

6. There are some slight mistakes in the manuscript. For example, the format of references, punctuation in the table should be unified:

Page 4：Fractal Dimension Index Analysis is used to analysis the fragmentation of urban land. The fractal dimension index can better indicate the fragmentation degree of urban land and the complexity of the irregular boundary of urban land patches [26].

Page 9：The evolution process of the urban land use spatial form in Feixi county was analyzed in conjunction with the growth mode of landscape pattern [27].

Page 11：The rapid expansion of urban land in Feixi county is related to the demand of its own development and the radiation and land demand of Hefei city. The land urbanization of Feixi county is carried out under the guidance of Hefei city and taking into account its own development.

Page 12：In the 1990s, China entered the climax of development zone construction, with development zones, industrial parks, and township industrial parks springing up all over the country [29](Cartier, 1990).

Page 15：In particular, county-level urbanization is much emphasized to facilitate urban-rural integration during this new wave of urban transformation ().

Page7: Table 1: Urban land (hm2).

Reply: Thanks very much. We have revised them.

Page 4：Fractal Dimension Index Analysis is used to analysis the fragmentation of urban land. The fractal dimension index can better indicate the fragmentation degree of urban land and the complexity of the irregular boundary of urban land patches.

Page 9：The evolution process of the urban land use spatial form in Feixi county was analyzed in conjunction with the growth mode of landscape pattern.

Page 11：The rapid expansion of urban land in Feixi county is related to the demand of its own development and the radiation and land demand of Hefei city. The land urbanization of Feixi county is carried out under the guidance of Hefei city and taking into account its own development.

Page 12：In the 1990s, China entered the climax of development zone construction, with development zones, industrial parks, and township industrial parks springing up all over the country(Cartier, 1990).

Page 15：In particular, county-level urbanization is much emphasized to facilitate urban-rural integration during this new wave of urban transformation.

Page7: Table 1: Urban land (hm2).

7. The spelling of this manuscript requires proofreading and I recommend English language editing by a native speaker.

Reply: Thanks very much. This manuscript has been proofread by a native speaker.

Thanks very much again. 

Best Wishes! 

Author

---

## [Decision Letter · Decision Letter 1]

22 Mar 2021

PONE-D-20-39602R1

The Characteristics of Spatial Expansion and Driving Forces of Land Urbanization in Counties in Central China: A Case Study of Feixi County in Hefei City

PLOS ONE

Dear Dr. Cao,

Thank you for submitting your manuscript to PLOS ONE. After careful consideration, we feel that it has merit but does not fully meet PLOS ONE’s publication criteria as it currently stands. Therefore, we invite you to submit a revised version of the manuscript that addresses the points raised during the review process.

The current version is much clearer and more informative. Thus I suggest to accept the manuscript after minor revision.

We look forward to receiving your revised manuscript.

Kind regards,

Mingxing Chen, Ph.D.

Academic Editor

PLOS ONE

Journal Requirements:

Reviewers' comments:

Reviewer's Responses to Questions

**Comments to the Author**

1. If the authors have adequately addressed your comments raised in a previous round of review and you feel that this manuscript is now acceptable for publication, you may indicate that here to bypass the “Comments to the Author” section, enter your conflict of interest statement in the “Confidential to Editor” section, and submit your "Accept" recommendation.

Reviewer #2: All comments have been addressed

Reviewer #3: All comments have been addressed

2. Is the manuscript technically sound, and do the data support the conclusions?

Reviewer #2: Partly

Reviewer #3: Yes

3. Has the statistical analysis been performed appropriately and rigorously? 

Reviewer #2: Yes

Reviewer #3: Yes

4. Have the authors made all data underlying the findings in their manuscript fully available?

Reviewer #2: Yes

Reviewer #3: Yes

5. Is the manuscript presented in an intelligible fashion and written in standard English?

Reviewer #2: Yes

Reviewer #3: Yes

6. Review Comments to the Author

Reviewer #2: I am pleased to read your revision manuscript. I noticed changes of the manuscript. A few issues are still suggested to be addressed.

（1） The theoretical framework and theoretical background of the manuscript need further improvement. I suggest the authors consider the following paper to upgrade the theoretical framework:

Li G , Sun S , Fang C . The varying driving forces of urban expansion in China: Insights from a spatial-temporal analysis[J]. Landscape and Urban Planning, 2018, 174:63-77.

Li M , Zhang G , Liu Y , et al. Determinants of Urban Expansion and Spatial Heterogeneity in China[J]. International Journal of Environmental Research and Public Health, 2019, 16(19):3706.

（2） The format of the chart in the manuscript is not standardized and unified. It is suggested to refer to the relevant paper format for modification.

（3） There are some slight problems in the manuscript. For example, whether is it appropriate to capitalize some words in the manuscript: “Keywords: land Urbanization; spatial Expansion; driving Forces; Feixi county”. “land Urbanization” in the manuscript.

I wish the authors good luck with the revision of the manuscript and the future research.

Reviewer #3: I agree with the comments of reviewers, and authors have revised the manscript carefully according to comments of previous round. It is appropriate to accept. Also, there are slight revision suggestions for authors before publication. First, it is recommend that increase the recent literature on new-type urbanization in introduction part. Second, based on the results of this research, add the discussions or policy suggestions for land urbanization of counties or cities in central China at the end of the research.

7. PLOS authors have the option to publish the peer review history of their article (what does this mean?). If published, this will include your full peer review and any attached files.

Reviewer #2: No

Reviewer #3: No

---

## [Author Response · Author response to Decision Letter 1]

8 May 2021

Dear Editor,

Thank you very much for the comments of the reviewers. We have carefully revised them. The following is our response:

Reviewer #2: I am pleased to read your revision manuscript. I noticed changes of the manuscript. A few issues are still suggested to be addressed.

（1） The theoretical framework and theoretical background of the manuscript need further improvement. I suggest the authors consider the following paper to upgrade the theoretical framework:

Li G, Sun S, Fang C. The varying driving forces of urban expansion in China: Insights from a spatial-temporal analysis[J]. Landscape and Urban Planning, 2018, 174:63-77.

Li M, Zhang G, Liu Y, et al. Determinants of Urban Expansion and Spatial Heterogeneity in China[J]. International Journal of Environmental Research and Public Health, 2019, 16(19):3706.

Reply: Thanks very much. 

the following paper has been added the the theoretical framework:

Li G, Sun S, Fang C. The varying driving forces of urban expansion in China: Insights from a spatial-temporal analysis[J]. Landscape and Urban Planning, 2018, 174:63-77.

Li M, Zhang G, Liu Y, et al. Determinants of Urban Expansion and Spatial Heterogeneity in China[J]. International Journal of Environmental Research and Public Health, 2019, 16(19):3706.

Li et al.(2018) point that multiple factors including socioeconomic, physical, proximity, accessibility, and neighborhood factors have driven urban expansion in China; Li et al.(2019) explored the spatiotemporal characteristics of Chinese urban expansion and adopted a geographically weighted regression (GWR) method to determine this spatial heterogeneity, The results revealed the spatial heterogeneity in the determinants of urban expansion, marketization was vital for urban expansion and had a stronger impact in the developed eastern and southern regions than in the less-developed northern and western regions, globalization and decentralization bi-directionally affected urban expansion.

（2） The format of the chart in the manuscript is not standardized and unified. It is suggested to refer to the relevant paper format for modification.

Reply: Thanks very much. We have standardized and unified the format of the chart.

（3） There are some slight problems in the manuscript. For example, whether is it appropriate to capitalize some words in the manuscript: “Keywords: land Urbanization; spatial Expansion; driving Forces; Feixi county”. “land Urbanization” in the manuscript.

I wish the authors good luck with the revision of the manuscript and the future research.

Reply: Thanks very much. We have modified, such as “Keywords: land urbanization; spatial expansion; driving forces; Feixi county”.

Reviewer #3: I agree with the comments of reviewers, and authors have revised the manscript carefully according to comments of previous round. It is appropriate to accept. Also, there are slight revision suggestions for authors before publication. First, it is recommend that increase the recent literature on new-type urbanization in introduction part.

Reply: Thanks very much. The following paper has been added the the theoretical framework:

Li G, Sun S, Fang C. The varying driving forces of urban expansion in China: Insights from a spatial-temporal analysis[J]. Landscape and Urban Planning, 2018, 174:63-77.

Li M, Zhang G, Liu Y, et al. Determinants of Urban Expansion and Spatial Heterogeneity in China[J]. International Journal of Environmental Research and Public Health, 2019, 16(19):3706.

Li et al.(2018) point that multiple factors including socioeconomic, physical, proximity, accessibility, and neighborhood factors have driven urban expansion in China; Li et al.(2019) explored the spatiotemporal characteristics of Chinese urban expansion and adopted a geographically weighted regression (GWR) method to determine this spatial heterogeneity, The results revealed the spatial heterogeneity in the determinants of urban expansion, marketization was vital for urban expansion and had a stronger impact in the developed eastern and southern regions than in the less-developed northern and western regions, globalization and decentralization bi-directionally affected urban expansion.

Second, based on the results of this research, add the discussions or policy suggestions for land urbanization of counties or cities in central China at the end of the research.

Reply: Thanks very much. From the perspective of better realizing the goal of land urbanization, according to the conclusions of this study, first, it is suggested to scientifically formulate and strictly implement the land use planning and strengthen the leading role of the planning in land urbanization; Second, for similar areas in central China, land supply should be effectively matched according to different development stages to improve land use efficiency.

Thanks very much again. 

Best Wishes! 

Author

---

## [Editor Report · Decision Letter 2]

14 May 2021

The Characteristics of Spatial Expansion and Driving Forces of Land Urbanization in Counties in Central China: A Case Study of Feixi County in Hefei City

PONE-D-20-39602R2

Dear Dr. Cao,

We’re pleased to inform you that your manuscript has been judged scientifically suitable for publication and will be formally accepted for publication once it meets all outstanding technical requirements.

Kind regards,

Mingxing Chen, Ph.D.

Academic Editor

PLOS ONE
---

## [Editor Report · Acceptance letter]

18 May 2021

PONE-D-20-39602R2 

The Characteristics of Spatial Expansion and Driving Forces of Land Urbanization in Counties in Central China: A Case Study of Feixi County in Hefei City 

Dear Dr. Cao:

I'm pleased to inform you that your manuscript has been deemed suitable for publication in PLOS ONE. Congratulations! Your manuscript is now with our production department. 

Kind regards, 

on behalf of

Prof. Mingxing Chen 

Academic Editor

PLOS ONE